# Hormonal Homologies between Canine Mammary Cancer and Human Breast Cancer in a Series of Cases

**DOI:** 10.3390/vetsci9080395

**Published:** 2022-07-29

**Authors:** Paloma Jimena de Andrés, Sara Cáceres, Juan Carlos Illera, Belén Crespo, Gema Silván, Felisbina Luisa Queiroga, Maria José Illera, Maria Dolores Pérez-Alenza, Laura Peña

**Affiliations:** 1Department of Animal Medicine, Surgery and Pathology, Veterinary School, Complutense University of Madrid, 28040 Madrid, Spain; mdpa@ucm.es (M.D.P.-A.); laurape@ucm.es (L.P.); 2Department of Animal Physiology, Veterinary School, Complutense University of Madrid, 28040 Madrid, Spain; sacacere@ucm.es (S.C.); jcillera@ucm.es (J.C.I.); belencre@ucm.es (B.C.); gsilvang@vet.ucm.es (G.S.); mjillera@vet.ucm.es (M.J.I.); 3Department of Veterinary Sciences, University of Trás-os-Montes and Alto Douro, 5000-801 Vila Real, Portugal; fqueirog@utad.pt; 4Animal and Veterinary Research Center (CECAV), University of Trás-os-Montes and Alto Douro, 5000-801 Vila Real, Portugal; 5Center for the Study of Animal Science (CECA), University of Porto, 4050-083 Porto, Portugal

**Keywords:** aromatase, breast cancer, canine mammary cancer, hormones, steroid receptors

## Abstract

**Simple Summary:**

There is worldwide interest in understanding the cancerous diseases that are causing increasing deaths in humans. In recent years, interest has grown in finding suitable models of different types of cancer in animals to lead the scientific community to a better understanding of the disease, in order to win the battle against cancer. The aim of this investigation was to compare breast cancer samples and canine mammary tumors from a hormonal point of view to validate the canine species as a model to study human breast cancer. There was a close similarity between premenopausal human breast cancer and canine mammary cancer in terms of hormonal receptors. In both species, all hormones assayed were increased in tumors compared to normal mammary gland samples. This research not only further supports canine mammary cancer as a spontaneous model for the study of human breast cancer but is also important in providing a deeper understanding of the hormonal pathogenesis of breast/mammary cancer in each independent species.

**Abstract:**

The validity of spontaneous canine mammary cancer (CMC) as a natural model for the study of human breast cancer (HBC) from a hormonal point of view has never been thoroughly investigated. In this study, we analyzed the immunohistochemical expression of aromatase (Arom) and steroid receptors [estrogen receptor α (ER α), estrogen receptor β (ER β), progesterone receptor (PR) and androgen receptor (AR)] and intratumor steroid hormone levels of 17β-estradiol (E2), estrone sulfate (SO4E1), progesterone (P4), androstenedione (A4), dehydroepiandrosterone (DHEA), and testosterone (T) in 78 samples of mammary cancer—51 human breast cancer (HBC) and 27 canine mammary cancer (CMC)—and corresponding controls. Frequency of tumors expressing Arom, ERβ, PR, and AR was similar in both species, whereas ERα+ tumors were less frequent in the canine species. There was a closer similarity between premenopausal HBC and CMC. In HBC and CMC, all hormones assayed were increased in tumors compared to control samples. Intratumor androgen levels were similar in the two species, although levels of progesterone and estrogens were higher in the HBC samples than the CMC samples. Statistical associations among Arom, receptors, and hormones analyzed suggest that the major hormonal influence in both species is estrogenic through the ER, being the α isoform predominant in the human samples. Our findings further support CMC as a spontaneous model for the study of HBC, especially premenopausal HBC, although several differences, such as the more prevalent ERα immunoexpression and higher intratumor levels of estrogens and P4 in HBC, should be taken into account in comparative hormonal studies.

## 1. Introduction

Human breast cancer (HBC) and canine mammary cancer (CMC) represent common neoplastic diseases that are still an important cause of mortality in women [1,2] and female dogs [3,4,5,6]. CMC is a spontaneous cancer that has been proposed as a model for the study of HBC [7,8,9,10,11,12,13,14,15,16,17], although some comparable aspects, including the hormonal profile, have not been evaluated in detail yet.

Although CMC is considered a natural model for the study of HBC [7,8,9,10,11,12,13,14,15,16,17], there are several minor characteristics in CMC with respect to HBC that should be considered depending on the type of comparative study. Histologically, the female dog has a high proportion of mixed tumors and tumors with myoepithelial cells [18] that should be avoided in comparative studies. Another interesting fact that is not usually mentioned is the absence of a menopausal status in the female dog, as well as differences regarding estrous cycles: healthy female dogs present only one or two estruses per year, which represents a low estrogenic influence compared to women [19]. Whether this hormonal variation in estrous cycles of both species may be reflected in the local hormonal status and receptors of mammary tumors remains obscure. The expression of estrogen receptor α (ERα) and progesterone receptor (PR) in canine mammary tumors has been postulated as one important characteristic to support the use of the canine model [7,8,9,10,20,21,22,23,24,25,26]. ERα and PR are routinely determined in HBC for prognostic and therapeutic purposes, but not in CMC, where the detection of ERα and PR is restricted to research. Several publications have demonstrated their expression in CMC, mostly in benign and low malignant tumors, and in some cases a relation to prognosis has been indicated [17,20,21,22,27,28,29,30,31,32,33]. ERβ expression in HBC [34,35,36,37,38] and CMC [28,32,39] is poorly researched and still controversial. ERβ is highly expressed in normal human breast tissue [34,35,37,40] and normal canine mammary glands [28,39]. Expression of the androgen receptor (AR) in breast cancer is even less studied, although in recent years research on AR expression has been increased for its prognostic value [41,42,43] and as a potential therapeutic target for triple-negative breast cancer [44,45,46,47,48,49,50,51]. Little is known about AR in CMC; the few published studies indicate a high prevalence of AR-positive tumors (85–92%) [13,28]. All these steroid receptors can trigger the growth action of their corresponding binding hormones that can be secreted locally. It is widely accepted that the mammary gland is a peripheral site for the production of steroid hormones [52]. The enzymes responsible for the synthesis of androgens, estrogens, and progestogens have been detected in normal and neoplastic mammary glands of several animals, including rats [53], goats [54], cows [55], dogs [11,56,57,58], and humans [59,60,61]. Among them, the expression of the enzyme cytochrome P450 aromatase (Arom, Cyp19a1, which catalyzes the conversion of androgens to estrogens in situ), has been analyzed in several studies in HBC [62,63,64,65] where the immunoreactivity of the enzyme is associated with its activity [63] and the immunoexpression of the ERα [62]. There is very little information regarding Arom in normal or neoplastic canine mammary glands [13,57,58,66].

The content of intratumor steroid hormones has not been researched thoroughly in HBC [67,68,69,70,71,72,73] or CMC [24,28,74], and the biological significance of in situ estrogen production with regard to the development and biological behavior of breast cancer remains controversial. Most studies indicate that intratumor estrogens derived from in situ aromatization could stimulate autocrine growth and function as mitogenic factors; therefore, these hormones might impart a growth advantage to the cancer cells, independent of estrogen serum concentrations [75,76,77]. Normal and neoplastic human breast tissues also contain and produce several forms of androgens, although this has not been studied extensively [68,69,70]. The few studies published on canine mammary neoplasms indicate that intratumor steroid hormones increase with malignancy [24,28,74].

In order to test the validity of canine mammary cancer as a model for human breast cancer from a hormonal point of view, the aims of this study were to compare in HBC and CMC samples the immunohistochemical expression of Arom and several hormone receptors (ERα, ERβ, PR, and AR), to analyze the intratumor steroid hormone concentrations (17β-estradiol, estrone sulfate, progesterone, androstenedione, dehydroepiandrosterone, and testosterone), and to study possible associations among these markers.

## 2. Materials and Methods

### 2.1. Sampling

#### 2.1.1. Human Breast Samples

For a year, fresh surgical breast samples of women with breast cancer clinically examined and treated in the San Carlos University Teaching Hospital were prospectively collected before chemotherapy, hormone therapy, or radiotherapy. Epidemiological and clinical data (age of the patient, menopausal status, tumor size, lymph-node status, and presence of metastases) were recorded. Clinical staging of the patients into the several clinical categories according to the TNM system was performed [78]. This system codes the extent of the primary tumor (T), regional lymph nodes (N), and distant metastases (M), and provides nine clinical stages based on T, N, and M. Patients were considered menopausal when they reported not having had any menses in the last 12 months. In total, 51 fresh surgical HBC samples from patients with a mean age of 63.3 years (range 33–89 years, 13 premenopausal and 38 menopausal) were prospectively collected (Table 1).

This study was conducted with the understanding and consent of every patient included and with the approval of the Ethical Committee of the San Carlos University Teaching Hospital (protocol code CEIC 19/545-E_BC).

#### 2.1.2. Canine Mammary Samples

During the same period, fresh surgical samples of 27 female dogs that presented at the Complutense Veterinary Teaching Hospital with mammary cancer (CMC) were obtained prior to any adjuvant therapy. Epidemiological and clinical information included age, ovariectomy status, tumor size, lymph-node status and presence of metastases. Clinical staging of animals into five clinical categories according to a modified WHO clinical staging system was performed [79]. The animals were of different breeds with a mean age of 10.2 (6–14) years. All the females were in anestrus or spayed at the time of sampling (Table 1). Fresh canine control mammary tissues (CC) were obtained by punch biopsies of the mammary glands performed on 8 healthy adult female beagle dogs in anestrus within the age range of 6–10 years and without any mammary or endocrine disorders. In this study, steps were taken at each stage of the experiment to avoid animal suffering. The study was performed with the consent of every owner and with the approval of the Animal Research Ethics Committee of the Complutense University of Madrid (PROEX 31/15).

### 2.2. Sample Processing

All the mammary samples (51 HBC, 10 HC, 27 CMC, 8 CC) were divided into 2 adjacent fragments: one portion was fixed in neutral formalin for histopathology and immunohistochemistry, and the other was frozen (−80 °C) and stored until the hormone contents were assayed.

#### 2.2.1. Histopathology and Immunohistochemistry

Formalin tissue samples were embedded in paraffin, cut into 3 µm sections and stained with hematoxylin–eosin. The histopathological diagnoses were done using the WHO’s classification system for HBC [80] and the most recent classification for CMC [18,81]. The histological grading was established using the Elston and Ellis system in human breast samples [82] and a similar system adapted for canine mammary cancer in the CMC group [83].

Immunohistochemical detection of Arom, ERα, ERβ, PR, and AR was performed in all mammary samples using antibodies with known reactivity in both species following the same methodology [13]. Immunohistochemistry of Arom was performed as follows: paraffin sections were placed in a PT module (Lab Vision Corporation, Fremont, CA, USA) containing an EDTA buffer solution (pH 8.0) (Master Diagnostica EDTA buffer solution MAD-004072R/D), heated for 20 min at 95 °C, and cooled to 60 °C. After this high-temperature antigen-retrieval protocol, slides were rinsed out in warm tap water and placed in an automated immunostaining device (Lab Vision Corporation, Fremont, CA, USA) for immunohistochemistry using a peroxidase detection system kit (MAD-021881QK, UltraVision Quanto-HRP). The slides were placed in a wet chamber at 4 °C overnight for incubation with the primary antibody (1/50, with known reactivity to canine Arom; ab35604, Abcam, Cambridge, UK). After immunostaining, the slides were hematoxylin-counterstained and permanently mounted with Depex. Corresponding negative control slides were done by replacing the primary antibody with a nonreactive antibody. Human placenta and canine ovary slides were used as positive controls. For the detection of the hormone receptors, immunohistochemistry was performed on dewaxed sections using the streptavidin–biotin–peroxidase method. After a high-temperature antigen-unmasking protocol (boiling slides in a pressure cooker for 2 min in buffer citrate, pH 6), endogenous peroxidase was blocked by immersion in 0.3% hydrogen peroxide for 15 min. Primary antibodies used were mouse monoclonal anti-ERα (1/20; C-311 Santa Cruz Biotechnology, Dallas, TX, USA), rabbit polyclonal anti-ERβ (1/200; 06-629, Upstate), mouse monoclonal anti-PR (1/40; 1A6, Novocastra, Newcastle upon Tyne, UK), and rabbit polyclonal anti-AR (1/25; N-Ab-2, Thermo Scientific, Waltham, MA, USA). All primary antibodies were incubated at 4 °C overnight. The secondary antibody for ERα and PR was horse anti-mouse IgG (diluted 1/400; BA2000, Vector), for ERβ goat anti-rabbit (1/300; E0432 Dako), and for AR swine anti-rabbit (1/200; E0353, Dako). All the secondary antibodies were incubated at room temperature for 1 h. After the secondary antibody, all the slides were subsequently incubated with streptavidin conjugated with peroxidase (1:400, 30 min at room temperature; 43-4323, Zymed, San Francisco, CA, USA). All washes and dilutions were done in Tris-buffered saline (0.1 M Tris base, 0.9% NaCl, pH 7.4). The slides were developed with a chromogen solution containing 25 g of 3–30 diaminobenzidine tetrachloride (D5059, Sigma, St. Louis, MO, USA), counterstained in hematoxylin (GH5-2-16, Sigma, St. Louis, MO, USA), washed in tap water, dehydrated, cleared in xylene, and mounted. Corresponding negative control slides were obtained by replacing the primary antibody with a nonreacting antibody on human or canine tissue. Normal canine uterus was used as a positive control for ERα, ERβ, and PR. When available in CMC, adjacent hyperplastic tissue was used as additional internal positive controls for these receptors. For AR, normal canine prostate was used as a positive control. When present, normal dermal sebaceous glands were used as additional internal positive controls for AR immunostaining.

Following the Allred score system [84], immunohistochemistry results of Arom (cytoplasmic immunolabeling) and hormone receptors (nuclear immunolabeling) were expressed as a total score (TS), calculated as the sum of the percentage of positive cells (PS) and the intensity of immunolabeling (IS). A sample was considered positive for Arom and hormone receptors when TS > 3.

#### 2.2.2. Assessment of Steroid Concentrations in Tumor Homogenate Samples

The analyses were performed according to previously published protocols [24,28]. After thawing, 0.5 g of the normal and neoplastic tissues were homogenized in 4 mL PBS (pH 7.2) and centrifuged (1372× *g*, at 4 °C for 20 min). The supernatants were collected and aliquoted individually (−30 °C) until the hormones were assayed.

17β-Estradiol (E2), estrone sulfate (SO4E1), progesterone (P4), androstenedione (A4), dehydroepiandrosterone (DHEA), and testosterone (T) levels of normal and neoplastic breast and mammary tissue homogenates were assayed by competitive EIA previously validated in our laboratory. Homogenate samples were prepared by diluting 10 µL of each homogenate in an assay buffer (1:2500 for DHEA, A4, T, and SO4E1 or 1:500 for E2), and then extracted with 2 mL diethyl ether (Sigma, St. Louis, MO, USA). One hundred microliters of the supernatant was evaporated under a nitrogen stream (Turbovap, ZIMARK, Hopkinton, MA, USA).

17β-Estradiol concentrations in tissue homogenates are expressed in pg/g. Levels of SO4E1, P4, A4, DHEA and T in tissue homogenates are expressed in ng/g.

### 2.3. Statistical Study

Analyses were performed using SPSS 19 and a conventional *p* < 0.05 level was used to define statistical significance.

Considering the absence of menopausal status in female dogs, three groups were established: premenopausal women (PM-HBC), menopausal women (M-HBC), and dogs (CMC). The categorical variables analyzed and the categories established were as follows: menopausal status in the women (premenopausal/menopausal); histological malignancy grade (I/II/III); Arom and steroid receptors (negative/positive). Pearson chi-squared statistical analysis was used to test the association among the different categorical variables within the groups. The numerical variables Allred TS (mean ± S.E) [84] and steroid concentrations (E2, SO4E1, P4, A4, DHEA and T) (mean ± S.E) were analyzed for their correlations (bivariate Pearson correlation test) and to find associations with the categorical variables (T tests).

## 3. Results

### 3.1. Histopathology

Histopathology of human and canine samples revealed several histological subtypes of carcinomas. The HBCs were diagnosed as follows: invasive ductal carcinomas (*n* = 38), mucinous carcinomas (*n* = 5), invasive lobular carcinomas (*n* = 4), invasive papillary carcinomas (*n* = 2), and tubular carcinomas (*n* = 2). CMCs were diagnosed as follows: tubular carcinoma (*n* = 7), tubulopapillary carcinoma (*n* = 5), solid carcinoma (*n* = 5), complex carcinoma (*n* = 3), mixed-type carcinoma (*n* = 3), adenosquamous carcinoma (*n* = 2), lipid-rich carcinoma (*n* = 1), and anaplastic carcinoma (*n* = 1) (Table 2).

### 3.2. Immunoexpression of Arom, ERα, ERβ, PR, and AR

Human breast and canine mammary controls were positive for all markers studied.

Immunolabeling of Arom was cytoplasmic in the neoplastic epithelial cells and heterogeneous within the same tumor section in both HBCs and CMCs. In the normal breast/mammary glands, Arom immunolabeling was weak and homogeneous. The rest of the markers studied (ERα, ERβ, PR, and AR) presented nuclear immunolabeling with variable heterogeneity: the heterogeneity of immunostaining across the same section was more evident in ERα and PR, and in CMC samples.

The immunoexpression of Arom and steroid receptors (ERα, ERβ, PR and AR) in the human and canine samples is shown in Figure 1.

In the HBC group, Arom+ tumors were more frequent in the menopausal group, although this association did not reach statistical significance (*p* = 0.15) (86.8% Arom+ in M-HBC vs. 69.2% of PM-HBC; 13.2% of Arom- in M-HBC vs. 30.8% of Arom- in PM-HBC). Arom immunolabeling was similar in HBC and CMC. All steroid receptors were more expressed in HBC than in CMC, specially ERα (TS, *p* < 0.001; % of positive tumors, *p* = 0.002) in M-HBC compared to CMC (Table 3).

### 3.3. Steroid Concentrations in Tumor Homogenate Samples

Hormone tissue levels of E2, SO4E1, P4, A4, and T were significantly higher in malignant samples than in normal breast/mammary samples in all the groups studied (PM-HBC, M-HBC, and CMC). Levels of DHEA were also higher in all groups, although the values did not reach significance in the PM-HBC group.

Steroid hormone concentrations in tumor homogenates are depicted in Table 4.

### 3.4. Statistical Associations among Aromatase, Receptors, and Steroid Hormones

#### 3.4.1. Arom Associations

In HBC tumors, Arom (−/+) immunoexpression was significantly associated with high ERα immunolabeling (TS and −/+) and AR (TS) as follows: 85.4% (35/41) of Arom+ tumors were ERα+ (*p* = 0.015); AR immunoexpression in Arom+ tumors was lower than in Arom- tumors (AR TS 6.44 ± 0.20 in Arom+ tumors vs. 7.60 ± 0.31 in Arom- tumors; *p* = 0.005). In addition, there was a positive statistical association between Arom (−/+) and DHEA (Arom+ tumors had higher levels of tissue DHEA than Arom- tumors; *p* = 0.016). In the premenopausal group, there was a significant statistical association between Arom (−/+) and tumor SO4E1 as follows: in the Arom+ samples, mean intratumor SO4E1 was higher than mean intratumor SO4E1 of the Arom− samples (*p* = 0.021). In the menopausal group, Pearson’s correlation was negative between Arom and AR TS (*p* = 0.036).

In the CMC group, Arom (−/+) was significantly associated with ERβ immunoexpression (−/+) (94.7%, 18/19 of Arom+ tumors were also ERβ+) (*p* = 0.031) and with PR (−/+) (94.7%, 18/19 of Arom+ tumors were PR+, whereas 75% (3/4) of the PR- tumors were Arom- (*p* = 0.031). In addition, Arom and ERβ TS were positively correlated (*p* = 0.018).

#### 3.4.2. Steroid Receptor Associations

In HBC, there was a strong positive association between ERα immunoexpression (TS and −/+) and PR immunoexpression (TS and −/+) (*p* values < 0.001). There was also a positive association between ERβ (TS and −/+) and AR (TS) (*p* < 0.001 and *p* = 0.003, respectively). On the other hand, ERβ (TS) was inversely associated with many hormonal contents as follows: P4 (*p* = 0.024), E2 (*p* = 0.032), and SO4E1 (*p* = 0.002). This negative association was also observed among AR (TS) and tissue contents of E2 and SO4E1 (*p* values < 0.001). With regard to PM-HBC, ERα and PR were also positively associated (TS, *p* = 0.010). Also in this group, PR+ tumors had lower content of T than PR- tumors (*p* = 0.020). Similarly, there was a positive association between ERα and PR in the M-HBC group. This group also showed a strong positive correlation between ERβ and AR (*p* < 0.001). Additionally, not only the ERβ but also the AR were negatively associated with the tumor content of SO4E1 (*p* = 0.007 and *p* = 0.042, respectively).

In the CMC group, several markers were positively associated: ERα/AR (*p* = 0.038), ERβ/PR (*p* = 0.002) and ERβ/AR (*p* = 0.036). Furthermore, the expression of several hormone receptors was negatively associated with hormones in the tumor homogenates: ERα/T (*p* = 0.022), ERα/DHEA (*p* = 0.014), ERβ/T (*p* = 0.022), and PR/SO4E1 (*p* = 0.005) and PR/T (*p* = 0.034). Another finding was that AR- tumors contained more A4 than AR+ tumors (*p* = 0.002).

#### 3.4.3. Associations among the Different Hormonal Tissue Concentrations

Regarding the correlations among the tumor hormone concentrations, there was a positive correlation of T with DHEA in PM-HBC group (*p* = 0.032). In CMC, T concentration was positively correlated with SO4E1 (*p* < 0.001) and A4 (*p* = 0.007). Also in the CMC group, A4 was positively correlated with DHEA (*p* = 0.017).

Homologies between human breast cancer and canine mammary cancer of the different markers and hormones analyzed are summarized in Table 5.

## 4. Discussion

Female dogs with mammary cancer are considered a spontaneous natural model for breast cancer, being especially useful in studies of new therapeutic approaches [7,9,10,11,85,86]. Nevertheless, slight differences between the tumors of the two species should be taken into consideration to make any comparison feasible. All the studies proposing the canine species as a model for the study of human breast cancer are based on neoplasms that occur spontaneously in the animal, and not induced as occurs in rodents, which makes them more reliable. Beside this, it is interesting to note that dogs live much longer than rodents and in the same environment as humans, both of which are advantageous for observing breast cancer kinetics and progression [8]. A limitation of the use of the spontaneous canine model for the study of human breast cancer is the lack of control over its appearance; however, this disease is the most frequent cancer in females of both the human and canine species, with increasing incidence in both species [87,88]. Most previous studies indicating the dog as an animal model for breast cancer have not made a proper comparison using samples of both species to verify the similarities between both types of cancers using the same methodology. An exception is the study of Uva et al. [85] that analyzed the gene expression of both CMC and HBC and normal mammary samples, finding a great degree of similarity in the perturbation of many cancer–related pathways. The present study assessed the extent to which the canine mammary tumors and human breast cancers are comparable in terms of hormonal background. To this end, the immunohistochemical expression of Arom and steroid receptors (ERα, ERβ, PR, and AR) and hormonal (E2, SO4E1, P4, A4, DHEA, and T) content in a series of tumors of both species, as well as the existing associations between them, were compared for the first time. Tumor content of steroid hormones in HBCs and CMCs has been little studied; therefore, the present study is also important in terms of understanding the hormonal pathogenesis of breast/mammary cancer in each species.

The proper hormonal comparison between CMC and HBC required dividing HBC into premenopausal (PM-HBC) and menopausal (M-HBC), given the differences in the estrous cycles among these three groups. In general, the immunohistochemical expression of Arom and the hormonal receptors studied showed evidence of a closer similarity between CMC and PM-HBC. Regarding the intratumor hormonal content, the three groups showed similar increases compared to the normal breast/mammary tissue.

The joint comparison of the immunohistochemical expression of steroid receptors between the two species has never been investigated. The cases included in this study, despite being collected prospectively, showed a similar percentage of lymphatic invasion and distant metastases in both species. In addition to this, the percentage of high-grade tumors in the human and the canine samples was similar. Therefore, the samples could be considered comparable from the clinical and histological point of view of malignancy. The immunoexpression of the receptors in both species is similar to previous results published separately [27,28,29,30,39,89,90,91,92]. In general, the human samples showed higher immunoexpression of the hormonal receptors studied; however, the percentages of positive tumors to most hormonal receptors and Arom were similar when comparing CMC with PM-HBC.

The higher immunoexpression of ERα (TS and −/+) in M-HBC was the major difference observed between HBCs and CMCs. In the HBC group, ERα was positively associated with Arom, as has been previously observed [62]. Moreover, in the human samples, we observed a negative association between ERβ and tumor content of E2 and SO4E1. These results may support the main action of the E2 via ERα in HBC. However, in CMC, given the low proportion of ERα+ tumors and the higher percentage of ERβ+ tumors (85.2% ERβ+ vs. 48.1% ERα+ in CMC), and that E2 levels increased in the CMC samples compared to CC, E2 may act mainly through the ERβ in mammary gland carcinogenesis in canine species, as has been previously suggested [28]. Moreover, in the canine samples there was a higher frequency of ERα−/RP+ tumors compared to the HBCs (44.4% in CMC vs. 16.6% in PM-HBC and 7.9% in M-HBC). Another important result is the positive association between ERα and PR found in the human samples, this result being in agreement with previous data [93,94]. Similarly, in the canine samples, this positive association was found between PR and ERβ. To date, there have been few studies on the expression of ERβ in HBC [34,35,36,37,38] and CMC [28,39]; therefore, the possible distinct role of ERβ requires further investigation.

Interestingly, similar percentages of PR+ and AR+ tumors in both species have been found, although the TS was higher in the human samples. In the M-HBC group, PR (TS) and AR (TS) were significantly increased compared to the CMC group (*p* = 0.003 and *p* < 0.001, respectively), and in the PM-HBC group, only AR (TS) was significantly increased (*p* = 0.001). On the other hand, the content of A4 in AR+ CMC tumors was lower than the content of A4 in the AR− tumors; therefore, it seems reasonable that A4 level decreases when the AR is present due to the conversion of A4 to T, which binds to AR.

Sex steroid formation in peripheral tissue is well documented in the human species, being the cytochrome P450 aromatase (Cyp19a1), an enzyme widely investigated in HBC [62,65,76,95,96,97,98]. Unfortunately, there is a paucity of information regarding Arom in CMC [13,57,58]. In this study, the increased tumor expression of Arom was associated with menopausal status in HBC which is related to the role of the mammary gland (normal or neoplastic) as a peripheral source of estrogens, especially in postmenopausal women [99].

Aromatase immunoexpression in the neoplastic cells was similar in both species. In HBC, it was associated with ERα status, an association observed previously by others [62,100] and, in CMC, with ERβ. Human breast cancer samples that showed immunoreactions to Arom were mainly ERα+, but showed low counts of AR TS. This suggests that when Arom is present, the local conversion of T to E2 could upregulate ERα and consequently downregulate the expression of AR. Similarly, in CMC, Arom+ tumors were also positive to ER, but in this species to the isoform β. This finding again supports the hypothesis of a major role of ERβ in the mammary carcinogenesis of canine species. Also in CMC, Arom was not inversely associated with AR, but it was positively associated with PR, as most of the Arom+ tumors were PR+. A different influence of progestogens upon human/canine mammary carcinogenesis should be taken into consideration, since the canine species is under major influence of progestogens, due to a longer period between consecutive estrous cycles in the female dog. The effect of progestogens in the mammary carcinogenesis of the dog has been previously indicated [101,102,103].

For the hormonal tumor content assay, fresh tissue was needed, and therefore it was necessary to prospectively collect samples from the limited number of cases included in the study. In spite of this, several interesting results were obtained. Both HBCs and CMCs contained higher amounts of all the hormones assayed compared with the respective normal control tissue of each species. These results are in agreement with previous studies on HBC [68,75,76,77] and CMC [24,28]. Nevertheless, when comparing the hormone concentrations in the tumors of humans and canines (Table 2), it is striking that estrogen (E2 and SOE1) and P4 levels were higher in HBCs than in CMCs, while the concentration of the androgens A4, DHEA, and T in both species was quite similar.

Another interesting finding was the dramatic increase in T observed in the CMC samples compared to their controls (CC) (44 times higher). Considering that there was a 2-point difference in AR TS between the species, AR being more frequent in human samples than canine samples, the role of T in the canine mammary carcinogenesis is still obscure. In the human samples, the hormone with a major increase was A4 for both premenopausal and postmenopausal HBC (12–16 times higher).

## 5. Conclusions

This study compares for the first time a series of CMC and HBC cases in terms of hormonal status using the same methodology. Interestingly, remarkable similarities have been found between CMC and HBC, especially PM-HBC, although some differences, such as the expression of ERβ, should be taken into account in comparative hormonal studies. Further comparative studies on the hormonal mechanisms of mammary carcinogenesis are needed to ensure the use of CMC as a good animal model for the study of HBC, including the hormonal perspective.

## Figures and Tables

**Figure 1 vetsci-09-00395-f001:**
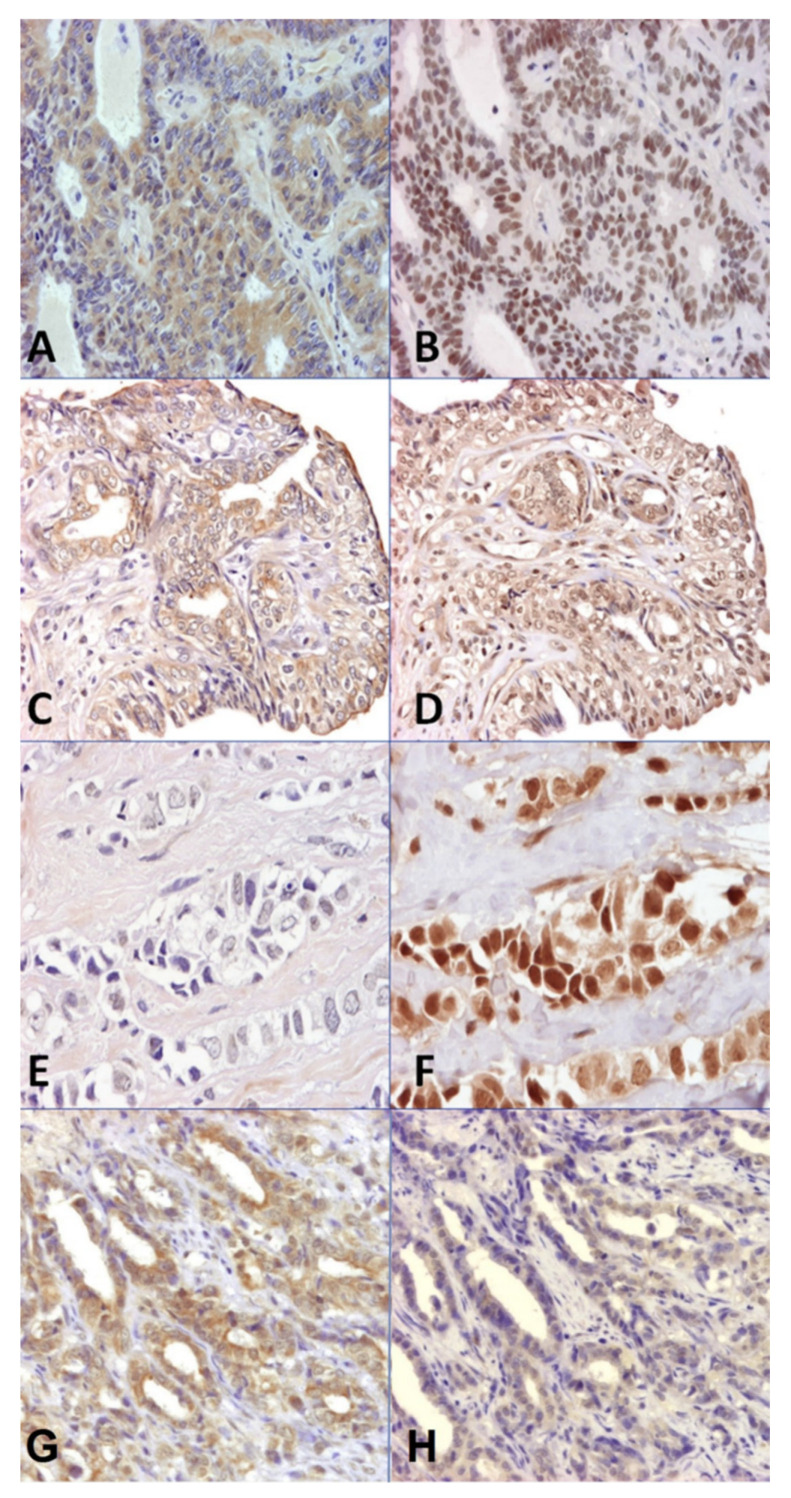
(**A**,**B**) HBC. Invasive papillary carcinoma. Original magnification (OM) ×10: (**A**) positive Arom immunostaining; (**B**) positive ERα immunostaining. (**C**,**D**) CMC. Tubulopapillary carcinoma. OM ×10: (**C**) positive Arom immunostaining; (**D**) positive ERβ immunostaining. (**E**,**F**) HBC. Invasive ductal carcinoma. OM ×40: (**E**) negative Arom immunostaining; (**F**) positive AR immunostaining. (**G**,**H**) CMC. Tubular carcinoma. OM ×10: (**G**) positive Arom immunostaining; (**H**) positive PR immunostaining.

**Table 1 vetsci-09-00395-t001:** Epidemiological and clinical information of the women and dogs with breast or mammary cancer prospectively collected for inclusion in the study.

	Women (*n* = 51)		Dogs (*n* = 27)
*Age at diagnosis (y)*		*Age at diagnosis (y)*	
Range	33–89	Range	6–14
Mean	63.3	Mean	10.2
*Menopausal status*		*Ovariectomy status*	
Premenopausal	13 (25.5%)	Intact (anestrus)	26 (96.3%)
Menopausal	38 (74.5%)	Spayed	1 (3.7%)
*Tumor size **		*Tumor size **	
T0	0	T1	11 (40.7%)
T1	32 (62.7%)	T2	5 (18.6%)
T2	16 (31.4%)	T3	11 (40.7%)
T3	3 (5.9%)		
T4	0		
*Lymphatic invasion ***		*Lymphatic invasion ***	
N0	30 (58.8%)	N0	20 (70.1%)
N1	16 (31.4%)	N1	7 (25.9%)
N2	5 (7.8%)		
Distant metastases †		*Distant metastases †*	
M0	49 (96%)	M0	22 (81.5%)
M1	2 (4%)	M1	5 (18.5%)
*Clinical stage ‡*		*Clinical stage ‡*	
0	0	I	11 (40.8%)
IA	21 (41.2%)	II	5 (18.5%)
IB	0	III	4 (14.8%)
IIA	13 (25.4%)	IV	2 (7.4%)
IIB	11 (21.6%)	V	5 (18.5%)
IIIA	4 (7.8%)		
IIIB	0		
IIIC	0		
IV	2 (4%)		

* Tumor size (in women T0: in situ, T1: tumor size < 2 cm, T2: tumor size 2–5 cm, T3 tumor size > 5 cm, T4; extension to chest wall or skin; in dogs T1: tumor size < 3 cm, T2: tumor size 3–5 cm, T3 tumor size > 5 cm). ** Lymphatic invasion (in women N0: no regional lymph-node metastases, N1: metastases to movable ipsilateral level I, II axillary lymph node, N2: metastases in ipsilateral level I, II axillary lymph nodes that are clinically fixed or matted; or in clinically detected ipsilateral internal mammary nodes in the absence of clinically evident axillary lymph-node metastases, N3: metastases in ipsilateral infraclavicular lymph nodes, metastases in ipsilateral internal mammary lymph nodes and axillary lymph nodes or metastases in ipsilateral supraclavicular lymph nodes; in dogs N0: absence of lymph-node involvement, N1: lymph-node involvement). † Distant metastases (M0: absence of metastases, M1: presence of distant metastases). ‡ Clinical stage in women: stage 0 (T0N0M0), stage IA (T1N0M0), stage IB (T0N1M0), stage IIA (T1N1M0 or T2N0M0), stage IIB (T2N1M0 or T3N0M0), stage IIIA (T0N2M0, T1N2M0, T2N2M0 or T3N1-2M0), stage IIIB (T4NanyM0), stage IIIC (TanyN3M0), and stage IV (TanyNanyM1); and in dogs: stage I (T1N0M0), stage II (T2N0M0), stage III (T3N0M0), stage IV (TanyN1M0), and stage V (TanyNanyM1). Additionally, fresh normal human control (HC) breast samples were obtained from 10 healthy adult women (6 premenopausal and 4 menopausal voluntary donors) without a history of breast or endocrine disorders and who were not using any hormone therapy (age range 23 to 65 years, mean 43.4 years). These patients had undergone surgical reduc-tion mammoplasties.

**Table 2 vetsci-09-00395-t002:** Histological subtypes and malignancy grade (HMG) of human breast cancer (HBC) and canine mammary cancer (CMC) samples prospectively included in the study.

	HBC (*n* = 51)		CMC (*n* = 27)
*Histological subtypes*		*Histological subtypes*	
Invasive ductal carcinoma	38 (74.6%)	Tubular carcinoma	7 (25.8%)
Mucinous carcinoma	5 (9.8%)	Tubulopapillary carcinoma	5 (18.6%)
Invasive lobular carcinoma	4 (7.8%)	Solid carcinoma	5 (18.6%)
Invasive papillary carcinoma	2 (3.9%)	Complex carcinoma	3 (11.1%)
Tubular carcinoma	2 (3.9%)	Mixed-type carcinoma	3 (11.1%)
		Adenosquamous carcinoma	2 (7.4%)
		Lipid-rich carcinoma	1 (3.7%)
		Anaplastic carcinoma	1 (3.7%)
*HMG*		*HMG*	
1	14 (27.4%)	1	13 (48.2%)
2	24 (47.1%)	2	7 (25.9%)
3	13 (25.5%)	3	7 (25.9%)

**Table 3 vetsci-09-00395-t003:** Immunoexpression of aromatase and steroid receptors in human breast cancer and canine mammary cancer samples.

		PM-HBC ^‡^ (a)	a vs. c *	M-HBC ^‡^ (b)	b vs. c *	CMC ^‡^ (c)
Arom (−/+) ^a^	Negative	4/13 (30.8%)	*p* = 0.941	5/38 (13.2%)	*p* = 0.102	8/27 (29.6%)
	Positive	9/13 (69.2%)		33/38 (86.8%)		19/27 (70.4%)
Arom ^b^	TS	4.77 ± 0.76	*p* = 0.696	5.45 ± 0.37	*p* = 0.133	4.37 ± 0.6
ERα (−/+) ^a^	Negative	4/13 (30.8%)	*p* = 0.209	6/38 (15.8%)	***p* = 0.002**	14/27 (51.9%)
	Positive	9/13 (69.2%)		32/38 (84.2%)		13/27 (48.1%)
ERα ^b^	TS	4.33 ± 0.96	*p* = 0.070	6.16 ± 0.47	***p* < 0.001**	2.44 ± 0.53
ERβ (−/+) ^a^	Negative	0/13 (0.0%)	*p* = 0.144	2/38 (5.3%)	*p* = 0.190	4/27 (14.8%)
	Positive	13/13 (100%)		36/38 (94.7%)		23/27 (85.2%)
ERβ ^b^	TS	6.85 ± 0.32	***p* = 0.003**	5.42 ± 0.31	*p* = 0.484	5.04 ± 0.48
PR (−/+) ^a^	Negative	2/13 (15.4%)	*p* = 0.962	4/38 (10.5%)	*p* = 0.604	4/27 (14.8%)
	Positive	11/13 (84.6%)		34/38 (89.5%)		23/27 (85.2%)
PR ^b^	TS	5.83 ± 0.89	*p* = 0.225	6.63 ± 0.42	***p* = 0.003**	4.74 ± 0.44
AR (−/+) ^a^	Negative	0/13 (0.0%)	*p* = 0.314	0/38 (0.0%)	*p* = 0.088	2/27 (7.4%)
	Positive	13/13 (100%)		38/38 (100%)		25/27 (92.6%)
AR ^b^	TS	6.92 ± 0.35	***p* = 0.001**	6.54 ± 0.22	***p* < 0.001**	0.37

^‡^ Premenopausal human breast cancer (PM-HBC), menopausal human breast cancer (M-HBC), and canine mammary cancer (CMC). * Numbers in bold denote significant differences (*p* < 0.05). ^a^ (−/+) Positive threshold when total score (TS) > 3. ^b^ Total Score calculated following the Allred score system.

**Table 4 vetsci-09-00395-t004:** Hormone concentrations (mean ± S.E.M) in breast/mammary tissue homogenates from women and dogs included in the study.

	Premenopausal Women		*p* Value *
	Normal Breast (*n* = 6)	PM-HBC (*n* = 13)	
E2 (pg/g)	237.56 ± 118.34	758.81 ± 136.38	**0.029**
SO4E1 (ng/g)	233.65 ± 41.31	1229.18 ± 153.96	**<0.001**
P4 (ng/g)	7.14 ± 0.57	66.04 ± 4.05	**<0.001**
A4 (ng/g)	19.19 ± 2.61	226.76 ± 12.57	**<0.001**
DHEA (ng/g)	161.46 ± 19.96	321.47 ± 50.51	0.052
T (ng/g)	9.77 ± 1.39	26.24 ± 2.74	**0.001**
	**Menopausal Women**		
	**Normal Breast** **(*n* = 4)**	**M-HBC** **(*n* = 38)**	***p* Value ***
E2 (pg/g)	73.86 ± 15.85	683.93 ± 88.23	**0.032**
SO4E1 (ng/g)	226.30 ± 25.84	1283.13 ± 140.97	**0.021**
P4 (ng/g)	12.91 ± 3.55	71.45 ± 2.56	**<0.001**
A4 (ng/g)	13.66 ± 1.68	213.79 ± 9.45	**<0.001**
DHEA (ng/g)	170.07 ± 5.59	328.55 ± 16.39	**<0.001**
T (ng/g)	8.46 ± 0.79	27.79 ± 1.43	**<0.001**
	**Female Dogs**		
	**Normal Mammary Gland** **(*n* = 8)**	**CMC** **(*n* = 30)**	***p* Value ***
E2 (pg/g)	89.74 ± 3.82	290.54 ± 31.8	**<0.001**
SO4E1 (ng/g)	76.51 ± 5.89	693.44 ± 85.93	**<0.001**
P4 (ng/g)	2.22 ± 0.12	8.69 ± 1.12	**<0.001**
A4 (ng/g)	11.13 ± 2.40	102.55 ± 10.69	**<0.001**
DHEA (ng/g)	28.24 ± 6.21	251.44 ± 13.80	**<0.001**
T (ng/g)	0.43 ± 0.07	20.15 ± 2.97	**<0.001**

** p* values in bold denote statistically significant differences between the control tissue and the corresponding breast or mammary cancer.

**Table 5 vetsci-09-00395-t005:** Comparison of the expression of aromatase, hormonal receptors, and local steroid hormone content between human breast cancer and canine mammary cancer samples.

	PM-HBC vs. CMC *	M-HBC vs. CMC *
*Immunohistochemistry* ^a^		
Aromatase +	++	+
ERα +	+	−
ERβ +	+	+
PR +	++	++
AR +	++	++
*Tumor steroid hormones content* vs. *controls* ^b^		
Levels of estrogens (E2, SO4) and P4	†	†
Levels of androgens (A4, DHEA and T)	‡	‡
*Hormonal associations*		
Positive correlation between androgens	+	−

* ++ = remarkably similar, + = similar and − = different, † = increase in hormonal content in tumor samples vs. control, ‡ = comparable increase in hormonal content in tumor samples vs. control. ^a^ Immunohistochemistry of Arom, ERα, ERβ, PR, and AR; percentages of positive tumors are considered. ^b^ 17β-estradiol (E2), estrone sulfate (SO4E1), progesterone (P4), androstenedione (A4), dehydroepiandrosterone (DHEA), and testosterone (T) analyzed by EIA.

## Data Availability

The data that support the findings of this study are available from the corresponding author upon reasonable request.

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
