# Peer review of "Hormonal Homologies between Canine Mammary Cancer and Human Breast Cancer in a Series of Cases"

_vetsci, 2022, doi:10.3390/vetsci9080395_

Round 1
Reviewer 1 Report
This study was well-designed and each of the scientific arms were well described and the results are set out clearly.
This paper was a delight to read.
Author Response
We are very thankful for this reviewer’s comments that encourage us to continue working and advancing in cancer research. We hope that you will find the small changes we have included appropriate by following the suggestions of the other reviewers.
Reviewer 2 Report
This work suggests that spontaneous canine mammary cancer (CMC) is a natural model for studying human breast cancer (HBC) from a hormonal point of view, especially in premenopausal HBC. However, they also found differences that should not be neglected while using canine mammary cancers and models to study human breast cancers. That should be better specified in the abstract and in the discussion. Regarding the abstract, for example, I suggest e similarities and differences instead of mixing them.
The results are clear, interesting, and important for future studies concerning human health. However, it would be interesting to have another approach, besides immunohistochemistry, such as Western-blot or even RT-qPCR (to address the gene expression profiles).
I point out some suggestions below to improve the manuscript.
- Abstract: I propose the following changes in two sentences:
A) “In this study, we analyzed the immunohistochemical expression of Aromatase (Arom) and steroid receptors and the intratumor steroid hormone levels of 17β-estradiol (E2), estrone sulfate (SO4E1), progesterone (P4), androstenedione (A4), dehydroepiandrosterone (DHEA), and testosterone (T) in 78 samples of mammary cancer [51 human breast cancers (HBC), 27 canine mammary cancers (CMC)] and the corresponding controls.”
2) “There was a closer similarity between premenopausal-HBC and CMC.”
Results: It would be interesting to have another approach, besides immunohistochemistry, such as Western-blot or even RT-qPCR, as explained above. More figures on immunohistochemistry would also be excellent.
Author Response
Please, see attachment

Reviewer 3 Report
The paper is about a study that intend to validate a spontaneous canine mammary cancer as a natural model for the study of human breast cancer from a hormonal point of view. Conceptually interesting, the methods used were well chosen and the results are consistent. However, considering the objectives of the study, the model has some limitations because, apart from divergencies related to the receptors expression, the spontaneous incidence of canine mammary cancer make the model difficult to apply. Nevertheless, the paper is interesting and should be published.
The authors should review the references and write them according to the journal rules.
Author Response
Please, see attachment

Reviewer 4 Report
This is an interesting manuscript focuses on the evaluation of canine breast cancer as a model of HBC. In general, the manuscript is well written and clear. The introduction provides enough background about the topic, the research methodology is adequate and sufficiently detailed and results are clear. Its deserves to be published. Just few comments/questions.
- I am wondering about the classification of the samples according to receptors expression. I can image that all luminal A and B BC.
- I am wondering. Did the authors found any diferences in terms of clinical stage of the disease?
Minor points:
- Samples calssification (section 3.1) I suggest including the information in this section (lines 207-213) in a table. I think in this case it would be more visual.
- I suggest including ethical approval numbers.
- I suggest also including a table summarizing the information of evaluated samples.
-Figure 1: Why do the images have different focuses? It should be included.
Author Response
Please, see attachment
